# Comparison of Metal-Based PZT and PMN–PT Energy Harvesters Fabricated by Aerosol Deposition Method

**DOI:** 10.3390/s21144747

**Published:** 2021-07-12

**Authors:** Chao-Ting Chen, Shun-Chiu Lin, Urška Trstenjak, Matjaž Spreitzer, Wen-Jong Wu

**Affiliations:** 1Department of Engineering Science and Ocean Engineering, National Taiwan University, Taipei 10617, Taiwan; ctchenx@ntumems.net; 2Nano-Electro-Mechanical-Systems Research Center, National Taiwan University, Taipei 10617, Taiwan; sclin@ntumems.net; 3Advanced Materials Department, Jožef Stefan Institute, 1000 Ljubljana, Slovenia; urska.trstenjak@ijs.si (U.T.); matjaz.spreitzer@ijs.si (M.S.)

**Keywords:** lead magnesium niobate–lead titanate (PMN–PT), lead zirconate titanate (PZT), energy harvesting, aerosol deposition, energy harvesting

## Abstract

In this study, polycrystalline lead magnesium niobate–lead titanate (PMN–PT) was explored as an alternative piezoelectric material, with a higher power density for energy harvesting (EH), and comprehensively compared to the widely used polycrystalline lead zirconate titanate (PZT). First, the size distribution and piezoelectric properties of PZT and PMN–PT raw powders and ceramics were compared. Thereafter, both materials were deposited on stainless-steel substrates as 10 μm thick films using the aerosol deposition method. The films were processed as {3–1}-mode cantilever-type EH devices using microelectromechanical systems. The films with different annealing temperatures were characterized by scanning electron microscopy, energy-dispersive X-ray spectroscopy, and dielectric behavior measurements. Furthermore, the mechanical and electrical properties of PMN–PT- and PZT-based devices were measured and compared. The PMN–PT-based devices showed a higher Young’s modulus and lower damping ratio. Owing to their higher figure of merit and lower piezoelectric voltage constant, they showed a higher power and lower voltage than the PZT-based devices. Finally, when poly-PMN–PT material was the active layer, the output power was enhanced by 26% at the 0.5 g acceleration level. Thus, these devices exhibited promising properties, meeting the high current and low voltage requirements in integrated circuit designs.

## 1. Introduction

With the advent of the Internet of Things (IoT), an increasing number of sensing units have been connected to the Internet. Although this advancement helps improve the quality of life, the sensing units consume substantial electricity. Battery-powered sensors are generally designed in an embedded form; therefore, replacing the batteries incurs additional cost, and the used batteries cause environmental pollution. However, the requirement of miniaturized devices led to the development of microelectromechanical systems (MEMSs) and semiconductor technology, which is aimed at lowering the power consumption of IoT sensing components. In contrast to eliminating power consumption, the compensation method creates self-powering abilities. Energy harvesting (EH), which allows a device to collect electrical energy from ambient energy sources, has been utilized in different studies [1]. The main goal of an EH device is to serve as the power source in batteryless systems. The EH-based self-powering system in a MEMS forms a one-package unit to fit the requirements of the IoT through an assembly with an interfacing circuit and sensing application device. A piezoelectric energy harvester (PEH) is easy to manufacture and is compatible with MEMS technology.

According to previous studies [2,3,4], the energy density of vibration sources is the highest available indoor energy at present and is practically inexhaustible. Owing to the inherent capability of piezoelectric materials, mechanical energy can be directly transformed into electrical energy. Hence, there is an urgent need for the miniaturization of the system. However, output power is proportional to the effective volume of the piezoelectric material [5]. The power consumption of these electronic components is gradually reduced to the sub-micron level, in which recent PEH devices cannot provide adequate power output. In addition to the advantage of MEMS compatibility, the design of a PEH is considerably simpler than that of other vibration energy harvesters, such as electromagnetic- and electrostatic-type energy harvesters [6,7,8]. To integrate them in the desired applications, high-performance PEH devices should be miniaturized to the centimeter scale. The miniaturization results in a decrease in power as the device scale decreases. To increase the energy output, a more effective mechanical-electrical structure should be redesigned or the piezoelectric quality should be improved through material modulation [9]. Another fundamental and more powerful solution is to allow for surpassing the power limitation of a PEH.

Researchers have increasingly attempted to improve the output performance of energy harvesters by optimizing the fabrication processes [10,11,12]. Over the past few years, lead zirconate titanate (PZT) [13,14,15] has been the most widely used piezoelectric material because of its excellent piezoelectric properties. To date, the energy output of energy harvesters gradually reached the upper limit of the chosen materials. In our previous research, power output was significantly enhanced, and the durability was improved by altering the EH substrate material [16]. Therefore, the introduction of a new material was considered the most promising method to overcome the limitation of output performance. Recently published studies [17,18,19] have focused on single-crystalline lead magnesium niobate–lead titanate (PMNPT) because of its high piezoelectric constant and electromechanical coupling factor [20]. Hence, it is a highly suitable next-generation piezoelectric material. However, the fabrication of epitaxial PMN–PT thin films is challenging due to interface mismatch. Accordingly, Baek et al. proposed a solution to this obstacle by introducing an SrRuO_3_/SrTiO_3_ buffer layer between the PMN–PT layer and the silicon substrate [21]. Simultaneously, they presented a cantilever-type actuator with a length of 34 μm and a thickness of 443 nm. Their results showed that the single-crystalline PMN–PT film had a higher figure of merit (FOM) for actuators and energy harvesters than other commercial piezoelectric materials. Although single-crystalline PMN–PT has a potential application in EH, the fabrication technology remains expensive and difficult to realize at the industrial production level [22,23]. In contrast, the aerosol deposition method (ADM) is more productive and demonstrates multi-product variability [24,25]. The deposition rate of the ADM is approximately several micrometers per hour, and this indicates that it is appropriate for industrial fabrication. Thus, the ADM was adopted in this study to fabricate poly-PZT and poly-PMN–PT energy harvesters. To quantify the piezoelectric properties, the piezoelectric constants of deposited PZT and PMN–PT thick films were measured in this study. However, various studies reported that the piezoelectric constant is highly related to the fabrication process and the stress from the substrate. Muralt investigated the relationship between the piezoelectric constant and clamping stress from a substrate, and reported that the piezoelectric constant of free piezoelectric film is twice that of the film clamped from the substrate [26]. Akedo and Lebedev reported that the piezoelectric constant *d*_31_ of aerosol deposited PZT film increased from 20–30 to 130 pC/N by increasing the annealing temperature from 500 to 600 °C. They also indicated that the piezoelectric constant improved 10-fold after optimizing the poling electric field and poling temperature [27,28].

In this study, poly-PZT and poly-PMN–PT were deposited on stainless-steel substrates. Thereafter, both thick films were fabricated in the 6 × 9 mm {3–1}-mode of cantilever-type PEH devices. The effect of the annealing temperature on both films was then analyzed. Subsequently, the mechanical and electrical properties of the poly-PZT film and poly-PMN–PT film were compared. Finally, the output performances of the different types of PEH devices based on the two considered materials were compared.

## 2. Materials and Methods

### 2.1. Fabrication Process

The geometric characteristics of the EH device are shown in Figure 1. The width and length of the cantilever were 6 and 9 mm, respectively. The fabrication process of the EH device is shown in Figure 2. A 60 μm stainless-steel material was used as the substrate, and served as the bottom electrode of the EH device because of its conductive properties. The fabrication process involves seven stages. First, the stainless-steel substrate was precleaned with a sulfuric acid and hydrogen peroxide solution. Subsequently, approximately 10 μm thick PZT and PMN–PT films were deposited via the ADM and patterned using a lift-off process. A probe-type surface analyzer (ET4000; Kosaka Laboratory Ltd., Tokyo, Japan) was used to scan the thickness and roughness of the deposited films. The results indicated that the film thickness and surface roughness of the PZT layer were approximately 10.0 and 0.41 μm, respectively. As for the PMN–PT film, the thickness and roughness of the deposited films were approximately 10.1 and 0.13 μm, respectively. The PZT and PMN–PT surface profiles indicated the high quality of the aerosol-deposited films. The lift-off process was used to pattern a 200 nm Pt/20 nm Ti top electrode that was deposited using an electron beam evaporator. Next, the stainless-steel substrate was wet-etched using an aqua regia solution to release the EH beam. The EH device was then annealed at the optimum temperature, followed by the bonding of the tungsten proof mass using an epoxy glue.

### 2.2. Experimental Setup

The experimental setup of the measurement is shown in Figure 3. The piezoelectric EH device was mounted on a shaker that provided a continuous vibration source with different frequencies for the device. The shaker was driven by a sinusoidal signal from a function generator and amplified with a power amplifier. The output signal from the EH device was connected to different impedances to estimate the output power under different load impedances. Simultaneously, an accelerometer (BRÜEL & KJÆR, Type 4381, Nærum, Denmark) was mounted on the shaker to calibrate the excited vibration level. The related output average power was calculated using Equation (1):(1)$$P=\frac{{\left(\frac{V_{p-p}}{2\sqrt{2}}\right)}^{2}}{R},$$
where *P* is the output power with load, *V_p-p_* is the peak–peak value of the output voltage with load, and *R* is the load resistor.

Moreover, to quantify the output performance of the PEH prepared from different piezoelectric materials, the characteristics related to FOM were measured. The FOM for the energy of the PEH is shown as Equation (2) [29,30]:(2)$$FOM=\frac{e_{31}{}^{2}}{\varepsilon_{r}}\propto \frac{{\left(d_{31}\cdot E\right)}^{2}}{\varepsilon_{r}},$$
where *d*_31_ is the piezoelectric constant, *E* is the Young’s modulus, and *ε_r_* is the relative permittivity.

## 3. Results and Discussion

### 3.1. Powder Indentification

In the ADM, the particle size of the powder highly influences the quality of the aerosol-deposited film [25,31]. Thus, powders of different particle sizes, fabricated with different ball milling times, were tested in the deposition process. Our previous study results showed that larger particles tend to cause a spray etching effect rather than film growth. In contrast to using powder with smaller particles, this technique slows the deposition rate and lowers the deposition efficiency. Therefore, the PZT and PMN–PT powders were ball-milled to a particle diameter of approximately 1 μm in this study. The particle size distribution of the two powders was subsequently detected using a laser diffraction particle size analyzer (CILAS 930e). The results shown in Figure 4 indicate that the mean particle sizes of PZT and PMN–PT were 1.3 and 0.97 μm, respectively.

Before proceeding to EH device fabrication, phase identification of the PZT and PMN–PT powders was performed using an X-ray diffractometer (Bruker D4 Endeavor). In the preparation of PMN–PT ceramics, the pyrochlore phase easily appeared during the fabrication process, and was difficult to remove. The pyrochlore phase forms in the absence of Pb-rich synthesis environment, and it reduces piezoelectric properties [32]. The X-ray diffraction (XRD) patterns of the PZT and PMN–PT powders with 2θ = 10°–70° are illustrated in Figure 5. The Cu Kα X-ray source at the wavelength of 0.154 nm was used for the measurement. The pattern clearly indicated that the PZT and PMNPT powders did not have the pyrochlore phase. The materials were fabricated as discs of diameter 25 mm and thickness 2.07 mm for calculating the material properties. The resulting permittivity (*ε_r_*), piezoelectric constant (*d*_31_), and coupling coefficient (*k_p_*) of PZT and PMN–PT ceramics were provided by the Eleceram Technology Co., Ltd. (Taoyuan, Taiwan) and are listed in Table 1 [33,34].

### 3.2. Film Characterization

The PZT and PMN–PT films on the stainless-steel substrates were approximately 10 μm thick. The deposited films were annealed from 350 to 650 °C for 3 h at an annealing rate of 2.5 °C/min. The surface morphology and elemental analysis of the annealed films were observed by scanning electron microscopy (SEM; Hitachi S4800) and energy-dispersive X-ray spectroscopy (EDS; Hitachi S4800), respectively. Figure 6 shows the SEM images of the PMN–PT film annealed in the temperature range of 550–650 °C at 20 °C intervals (PZT film as shown in Appendix A). For the surface morphology, several exaggerated grains anchored on the surface were observed for the film annealed above 630 °C. The EDS analysis was performed to investigate the composition of the as-deposited and annealed PMN–PT films. Table 2 reveals the presence of iron and chromium and the lack of magnesium in the 650 °C annealed PMN–PT film. With an increase in the annealing temperature to up to 600 °C, the atomic composition of the film remained the same. However, when the annealing temperature was increased to 650 °C, a weak signal of Cr and Fe could be detected on the PMN–PT film. In addition, the EDS results indicated a very high intensity of Cr and Pb signals on the exaggerated grains with angular shape. However, as the ratio of the second phase on the film was relatively low, there was no second-phase signal in the XRD patterns of PMN–PT and PZT film annealed at 550 to 650 °C as shown in Figure 7. A similar phenomenon of the second phase, originating from the reaction of the piezoelectric material and stainless steel, was observed in a previous study [35]. Fe, Ni, and Cr signals were observed from the EDS analysis of the PZT film treated above 600 °C, leading to the dielectric losses.

The dielectric measurements showed that the 650 °C annealed PMN–PT film was conductive with poor piezoelectric properties. When the annealing temperature was decreased to 600 °C, the dielectric loss decreased to 0.07, and the permittivity increased to 426 at 1 kHz (Figure 7). The XRD spectra of the annealed PZT and PMN–PT films are shown in Figure 5. The films were phase pure, whereas the major peak in the PZT and PMN–PT powders shifted by 0.33° and 0.01° toward higher 2 values, respectively, which indicated that compressive residual stress remained in the piezoelectric layer via aerosol deposition.

### 3.3. Young’s Modulus and Hardness

The hardness corresponding to density is a quality factor for ceramic materials [36]. Thus, a higher density of the same ceramic material system indicates a greater hardness. The Young’s modulus and hardness of the annealed films were measured using a nanoindenter (Hysitron TI 950 TriboIndenter). A 200 nm indentation depth (<10% of film thickness) was adopted to avoid or limit the effect of the stainless-steel substrate [37,38]. Table 3 shows the comparison of the Young’s modulus and hardness of the deposited PZT and PMN–PT films. In addition, the characteristics of the PZT and PMN–PT single crystals were also included in the comparison. The Young’s modulus and hardness of the deposited PZT and PMN-PT film in this study are comparable with those of previously reported aerosol-deposited piezoelectric film; this confirmed the high quality of the deposited films prepared in this study. In addition, the higher Young’s modulus of the deposited PMN–PT film than that of the deposited PZT film improved the FOM.

### 3.4. Output Performance

Figure 8 shows the output voltage and power versus various load impedances under 0.5 g of excitation acceleration level. The output power was calculated using Equation (1). The maximum output voltage of the PZT-based device under open-circuit conditions was 13.6 V_p–p_ at a resonant frequency of 101.6 Hz, with the 0.5 g acceleration level. Conversely, the maximum output voltage of the PMN-PT-based devices was 10.8 V_p–p_ at a resonant frequency of 98 Hz with the 0.5 g acceleration level. The corresponding output voltages with the optimal loads of the PZT- and PMN–PT-based devices were 9.3 and 7.7 V_p–p_, respectively (Figure 9). The maximum output power of the PZT- and PMN–PT-based devices were 71.8 and 90.4 μW, respectively. A lower output voltage and higher output power, that is, a higher output current, could be observed from the PMN-PT-based devices with respect to the PZT-based devices. For the integrated circuit design, a down-scaling circuit component was used to avoid electric loss and to apply to wearable devices. The circuit components, particularly for miniaturized integrated circuit systems, cannot sustain the higher voltage provided by the piezoelectric transducer. Consequently, introducing an alternative material with a higher output current but lower output voltage is necessary and promising.

The resonance frequencies of the PZT- and PMN–PT-based EH devices were approximately 101.6 and 98.0 Hz, respectively. The difference in the resonance frequency was caused by the varying Young’s modulus. The well-known quality factor was determined by the resonant frequency over the full width at the half maximum. During the measurement, both devices were excited at a low acceleration level to avoid nonlinear behaviors. The quality factors of the PZT- and PMN–PT-based devices were 72.8 and 147.4, respectively. The damping ratio can be expressed as the reciprocal of two quality factors. The quality factor can be obtained by calculating the resonant frequency over the full width at the half maximum (FWHM) of the output voltage. The coupling coefficient $k_{e}^{2}$ was determined by measuring the open-circuit (ω_oc_) and short-circuit (ω_sc_) resonant frequencies [42]. To evaluate the piezoelectric constant of PZT and PMN–PT, a beam with a specific shape and laminar structure was fabricated [43]. The piezoelectric constant is strongly related to the fabrication process and poling condition [28,31]. To obtain a reliable comparison between the PZT and PMN-PT films, both films were annealed at 600 °C and poled at the highest electric field that they could sustain owing to the difference in dielectric strength. Therefore, PZT and PMN–PT were heated to 150 °C and poled with electric fields at 15 and 30 V/μm, respectively. The piezoelectric constants (*d*_31_) of PZT and PMN–PT were −17.8 and −26.9 pC/N, respectively, which were lower than that of the bulk material. The value is similar to the piezoelectric constant of a 30 μm thick aerosol-deposited PZT film, as reported by Akedo [27]. With the same fabrication condition, the PMN–PT-based device exhibited excellent piezoelectric properties with a high piezoelectric constant and Young’s modulus, but with a high permittivity. The coupling coefficients of the PZT-and PMN–PT-based devices were 0.10 and 0.11, respectively. Although the PMN–PT material has numerous excellent properties, the PMN–PT film presented a larger dielectric loss. Consequently, the power of the PMN–PT-based EH devices is 1.26 times larger than that of the PZT-based devices. A comparison of the PZT- and PMN–PT-based EH devices is presented in Table 4.

Because of the excellent piezoelectric properties of PMN–PT, the PMN–PT- and PZT-based energy harvesters are increasingly comparable. Table 5 shows a comparison of the output performance of the recently reported PZT- and PMN–PT-based energy harvesters on a metal substrate. Although the fabrication technology of the proposed PMN–PT film is not well-developed, the energy density is better than that reported previously [19,44,45,46,47]. We expect the output performance of the PMN–PT-based energy harvester can be improved by optimizing the fabrication process, including the deposition parameters, poling condition, and annealing profile.

## 4. Conclusions

In this study, we successfully deposited poly-PZT and poly-PMN–PT materials on stainless-steel substrates using the aerosol deposition method. The 10 μm thick PZT and PMN–PT films were fabricated as 6 × 9 mm {3–1}-mode cantilever-type EH devices. The single-phase PZT and PMN–PT devices were obtained and characterized considering the 600 °C annealing temperature. Regarding the mechanical properties, the PMN–PT film showed a higher Young’s modulus and hardness, indicating its potential in EH applications. The PMN–PT-based devices also showed higher quality factors and coupling coefficients, but they also exhibited higher unfavorable dielectric losses. Finally, the poly-PMN–PT-based EH devices produced an output power of 90.4 μW at an acceleration level of 0.5 g. The typical PZT-based EH device produced an output power of 71.8 μW at the same active volume and excitation level. In summary, we introduced an alternative piezoelectric material, that is, polycrystalline PMN–PT, as a PEH solution that is more suitable for integrated circuit components owing to its low voltage output. Poly-PMN–PT had a high piezoelectric constant and FOM compared to the widely used PZT material.

## Figures and Tables

**Figure 1 sensors-21-04747-f001:**
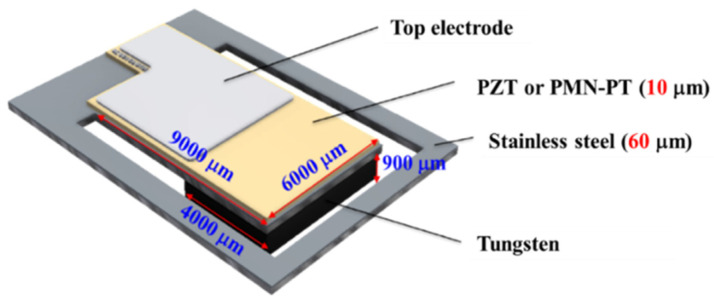
Illustration of the PMN–PT- or PZT-based micro-energy harvester with the geometric parameters.

**Figure 2 sensors-21-04747-f002:**
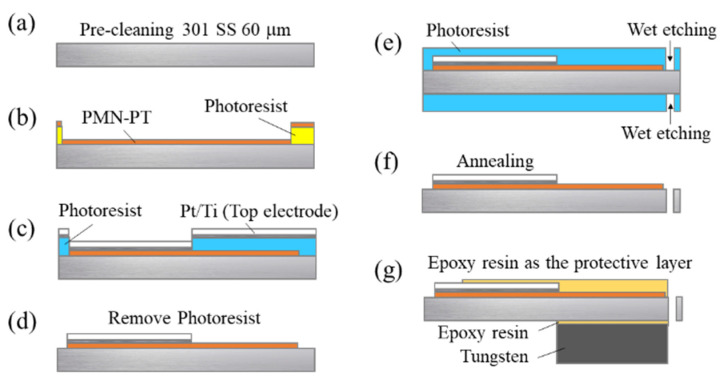
Schematic diagram of the fabrication process based on a stainless-steel substrate: (**a**) substrate precleaning, (**b**) PZT or PMN–PT thick-film deposition and patterning, (**c**) top electrode deposition, (**d**) lift-off process, (**e**) beam shape etching, (**f**) PZT or PMN–PT film annealing, and (**g**) protective layer coating and bonding of a proof mass.

**Figure 3 sensors-21-04747-f003:**
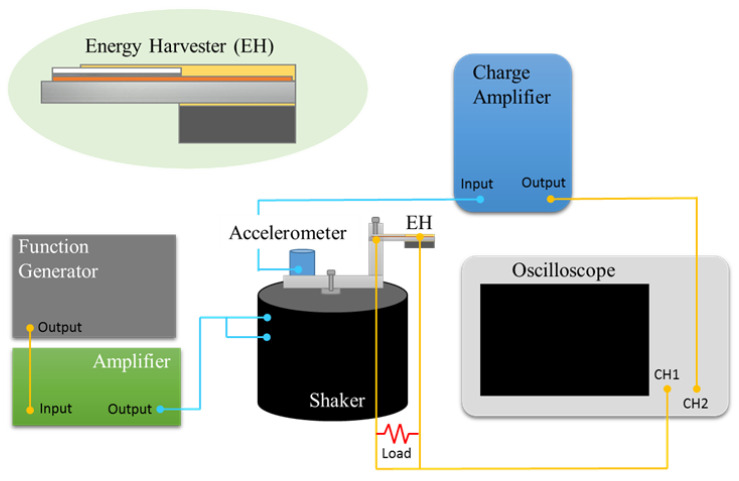
Schematic illustration of the experimental setup for the piezoelectric MEMS generator.

**Figure 4 sensors-21-04747-f004:**
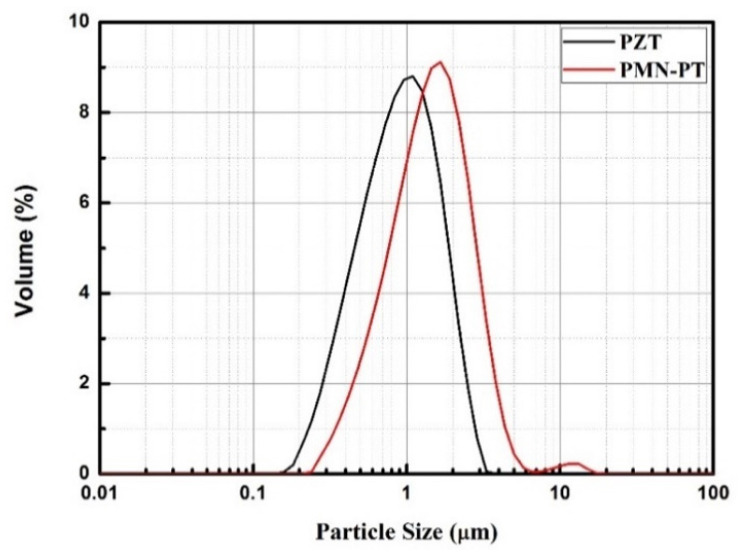
Particle size distribution for the PZT and PMN–PT powders.

**Figure 5 sensors-21-04747-f005:**
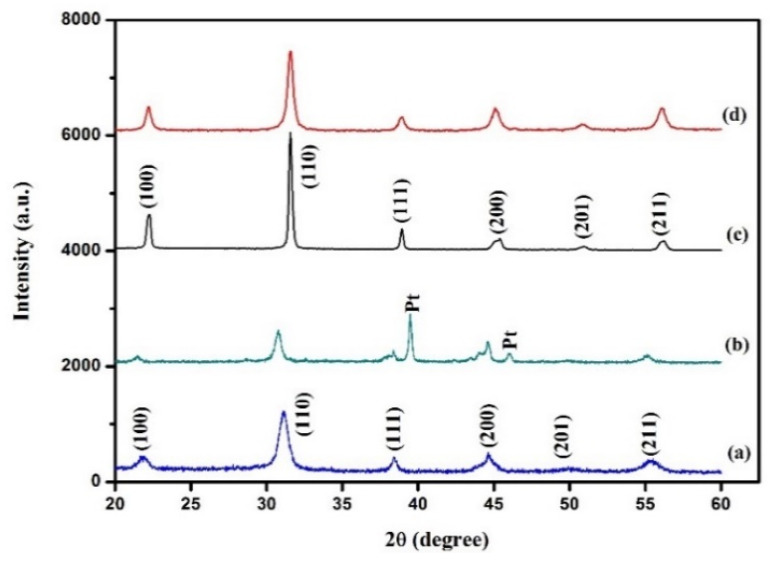
XRD patterns of the (**a**) PZT powder, (**b**) 600 °C annealed PZT film, (**c**) PMN–PT powder, and (**d**) 600 °C annealed PMN–PT film.

**Figure 6 sensors-21-04747-f006:**
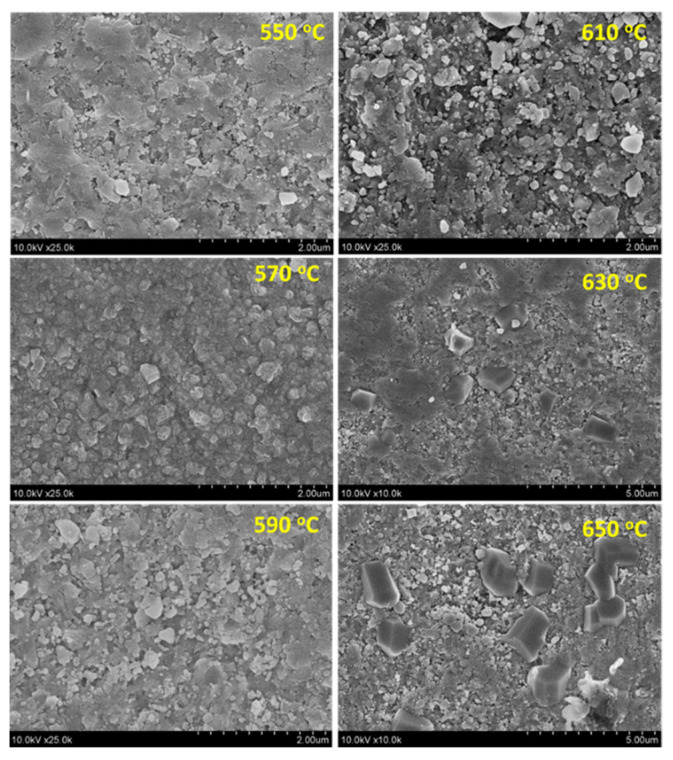
SEM micrographs of the PMN–PT films annealed in the temperature range of 550–650 °C.

**Figure 7 sensors-21-04747-f007:**
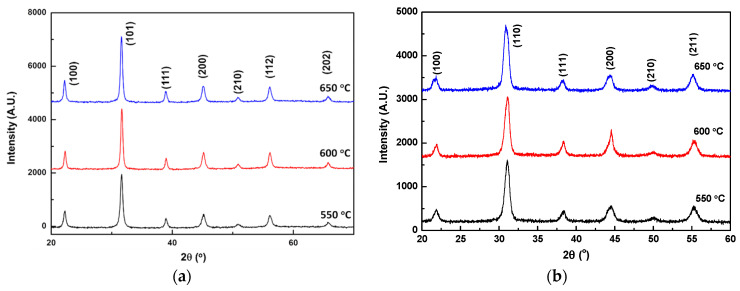
XRD patterns of the (**a**) PMN–PT and (**b**) PZT films annealed under 550, 600, and 650 °C.

**Figure 8 sensors-21-04747-f008:**
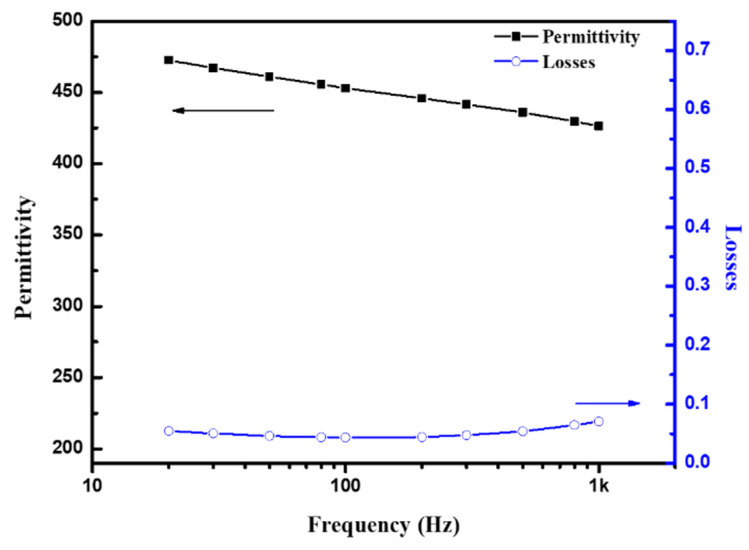
Frequency dispersion of permittivity and loss tangent of the 600 °C annealed PMN–PT film.

**Figure 9 sensors-21-04747-f009:**
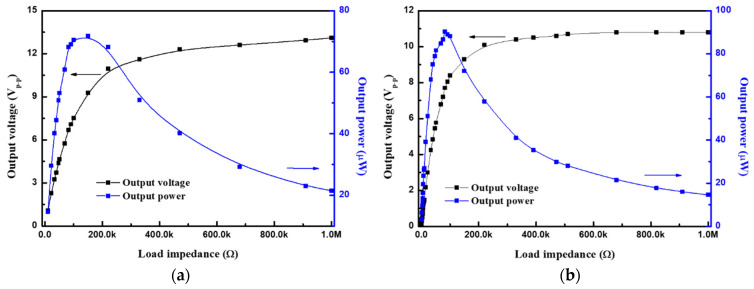
(**a**) PZT and (**b**) PMN–PT device electrical output vs. load impedance at the 0.5 g acceleration level.

**Table 1 sensors-21-04747-t001:** Comparison between PZT and PMN–PT.

Material	Permittivity	Piezoelectric Constant, *d*_3__1_ (pC/N)	Coupling Coefficient, *k_p_*
Poly-PMN–PT	3621	−250	0.69
Poly-PZT	1300	−130	0.6

**Table 2 sensors-21-04747-t002:** Atomic composition of the as-deposited and annealed PMN–PT films.

Material/Film Type	Pb (%)	Mg (%)	Nb (%)	Ti (%)	Cr (%)	Fe (%)
Stainless steel	-	-	-	-	21	71
Ceramic	47	8	25	20		
As-deposited film	45	9	26	19	-	-
600 °C annealed film	46	14	22	18	-	-
650 °C annealed film	39	6	28	18	2	8
Second-phase grain on 650 °C annealed film	55	0	7	2	29	5

**Table 3 sensors-21-04747-t003:** Comparison of the Young’s modulus and hardness of the PZT and PMN–PT materials.

Material	Young’s Modulus (GPa)	Hardness (GPa)
PZT film [39,40]	80	5–8
PMN–PT single crystal [41]	126.81	5.59
Poly-PMN–PT sheet [33]	70 ± 15	-
Poly-PZT (thick film) (this study)	90.75	5.9
Poly-PMN–PT (thick film) (this study)	86.47	5.59

**Table 4 sensors-21-04747-t004:** Comparison of the parameters of the PZT and PMN–PT materials.

Parameter	PZT	PMN–PT
Thickness (μm)	10	10
Resonant frequency (Hz)	101.6	98
Quality factor	72.8	147.4
Damping ratio	0.007	0.003
Capacitance (nF)	9.8	20.5
Permittivity, *ε_r_*	309	426
Piezoelectric coefficient, *d*_31_ ($\mathrm{pC}\cdot N^{-1}$)	−17.8	−26.9
Coupling coefficient, $k_{e}^{2}$	0.10	0.11
Dielectric loss (%)	1.5	3
Normalized FOM ${\left(C\cdot m^{-2}\right)}^{2}$	1	1.5
Output power (μW)	71.8	90.4
Output voltage with load (V_P–P_)	9.3	7.7
Optimum impedance (kΩ)	150	82

**Table 5 sensors-21-04747-t005:** Previously reported PZT- and PMN–PT-based energy harvesters with a metal substrate.

Study	Piezoelectric Material	Substrate Material	Dimension (mm^3^)	Acceleration (g)	Frequency (Hz)	Avg. Output Power (μW)	Energy Density (μW cm^−3^ Hz^−1^)
Morimoto [44]	PZT	Steel	4.9	0.5	126	5.3	34.5
Tsujiura [45]	PZT	Steel	17.1	0.1	143	1.8	73.6
Yang [19]	PZT	Aluminum	486	0.3	151	150	22.7
Ayse [46]	PZT	Brass	-	0.2	34	1.1	-
Gibus [47]	PZT	Steel	4500	0.02	32.1	0.6	10.2
This work	PZT	Steel	25.4	0.5	101.6	71.8	145.7
Yang [19]	PMN–PT*	Aluminum	486	0.3	159	250	35.9
Ayse [46]	PMN–PT*	Brass	-	0.2	35	1.2	-
Gibus [47]	PMN–PT*	Steel	3750	0.02	29.1	0.4	12.3
This work	PMN–PT	Steel	25.4	0.5	98.0	90.4	111.6

## Data Availability

The data presented in this study are available on request from the corresponding author.

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
