# Peer review of "Comparison of Metal-Based PZT and PMN–PT Energy Harvesters Fabricated by Aerosol Deposition Method"

_sensors, 2021, doi:10.3390/s21144747_

Round 1
Reviewer 1 Report
This manuscript fabricated and studied the properties of PMN-PT and PZT films and their performance as a EH device. A comprehensive comparison has been made and as a conclusion, the authors claim PMN-PT can be used as an alternative solution for PEH device. The motivation and background information have been clearly introduced. The fabrication methods are carefully described and the samples are carefully characterized. The evidences shown in this manuscript are convincing and their results are helpful for developing new energy harvesting devices. Therefore, I recommend this current paper for publication after minor revision.
Here are some minor comments:
- The chemical composition between as-grown and annealed film are different. Can the authors explain the reason and make comments about its possible consequence on their conclusion.
- In figure.8, the label on the right axis should be "output power" instead of "output powder".
Reviewer 2 Report
This study used aerosol deposition method to fabricate the MEMS energy harvester both by PMN–PT and PZT. The manufacturing processing and the microstructure evaluation are clearly presented. The performance in data of Young's modulus, electric power and voltage is consistent with the description with abstract. The reviewer suggests that the measuring method of damping ratio should be illustrated in detail. The comment or improved method can be provided after the voltage and power result in fewer than 20% difference, as shown in Fig. 8.
Reviewer 3 Report
I would like to have some comments for the manuscript as the attached file.
